# Good quality end-of life care for people with an intellectual disability: A critical interpretive synthesis protocol

**Margaret Haigh**[1]*, **Mary McCarron**[1], **Philip McCallion**[2], **Pavithra Pavithra**[1], **Martin McMahon**[1]

**1** School of Nursing & Midwifery, The University of Dublin, Trinity College Dublin, Ireland, **2** School of Social Work, College of Public Health, Temple University, Philadelphia, Pennsylvania, United States of America

* haighm@tcd.ie

**Data Availability Statement:** No datasets were generated or analysed during the current study. All

## Abstract

People with an intellectual disability face inequality accessing healthcare that extends to end-of-life care. Furthermore, end-of-life care for people with an intellectual disability is more complex than that for the general population. This protocol sets out how our review will explore the available evidence and identify the key characteristics of care that are required for people with an intellectual disability to have a good end-of-life experience. A critical interpretive synthesis approach will be adopted, combining some elements of a systematic review with interpretive synthesis. This approach has been selected as it allows a diverse body of literature to be considered. Furthermore, it is compatible with theory generation allowing new insights to emerge in an iterative process. Electronic databases and grey literature sources will be searched using pre-defined search terms. Following initial title and abstract screening, eligible papers will undergo full text screening. Papers that are deemed to be 'fatally flawed' will be excluded and remaining papers appraised for relevance. Resultant papers will be included in the sampling frame from which data extraction will occur. An existing framework outlining key factors in the provision of excellent end-of-life care to people with an intellectual disability will be used to support data extraction and synthesis. Data extracted will be integrated into a synthesising argument in the form of a theoretical framework. This will identify the key characteristics in the provision of care that are required for people with an intellectual disability to have a good end-of-life experience. The developing theoretical framework will guide further selection of relevant literature to fill any conceptual gaps until saturation is reached. Our review will add to the existing body of evidence, shedding light on the key factors necessary in the provision of care to ensure that people with an intellectual disability have a good end-of-life experience.

## Introduction

An intellectual disability is characterised by significant limitations in both intellectual functioning (i.e. intelligence) and adaptive behaviour (i.e. conceptual, social, and practical skills).

relevant data from this study will be made available upon study completion.

**Funding:** This work is partly funded by the Health Research Board - 'IDS-TILDA-2021-001'. Funding was awarded to MMcC. The funders had no role in study design, data collection and analysis, decision to publish, or preparation of the manuscript. https://www.hrb.ie/ No other funding was received.

**Competing interests:** The authors have declared that no competing interests exist.

Generally, a score of less than 75 in an intelligence quotient test is indicative of significant limitation in intellectual functioning [1]. Based on the findings of a meta-analysis of population based studies drawn from low and middle income countries as well as developed countries [2], it was suggested that approximately 1% of the population has an intellectual disability.

People with an intellectual disability are more likely to have poorer health than those without an intellectual disability [3]. In addition, death is reported to be earlier by around 20 years than for the general population [4]. While the poorer health of people with an intellectual disability can be attributed in part to greater prevalence rates of health problems, with gastrointestinal, endocrine and neurological diseases prevailing [5], people with an intellectual disability face other significant challenges making access to health care more difficult [3]. Barriers can result from communication difficulties [6], stigmatising attitudes among health professionals towards people with an intellectual disability [7] and inadequate knowledge about the health needs of people with an intellectual disability leading to diagnostic overshadowing and ultimately delayed diagnosis and treatment [8].

Despite the complex health issues and disadvantages experienced, life expectancy in people with an intellectual disability has increased in recent decades. This has been attributed to improved health care, including screening and other initiatives addressing lifestyle behaviours and health risks [4]. The increased life expectancy for people with an intellectual disability results in a growing need for end-of-life care for this ageing population [9]. As with the general population, compassionate and person-centred care is required [10].

Nevertheless, the literature confirms that providing end-of-life care for people with an intellectual disability is more complex than that for the general population. Many health care professionals report feeling inadequately trained to provide good end-of-life care to people with an intellectual disability [11] and a lack of understanding of the respective role of palliative care professionals and intellectual disability staff is reported [10]. Moreover, the inequality in access to healthcare for people with an intellectual disability extends to end-of-life care [12–14]. Though guidance is emerging regarding good practice in palliative care provision for people with an intellectual disability [15–17], there is a need for a comprehensive, systematic, well-organised theoretical framework, based on an appraisal of the literature, to support the implementation of best practice in this area.

The objective of this review will be to develop a theoretical framework outlining the key characteristics in the provision of care that are required for people with an intellectual disability to have a good end-of-life experience. The aim of this protocol is to set out the process for this review which will be conducted using a Critical Interpretive Synthesis (CIS) approach.

## Methods

The literature review will adopt a CIS approach based on the original review carried out by Dixon-Woods et al. [18]. This decision was informed by the findings of an initial review of the literature in relation to the end-of-life experiences of people with an intellectual disability which established that the body of evidence is similarly diverse and complex. This was a preliminary review carried out by the lead author and involved semi-structured searches of electronic databases. Discussions were also held with members of the research team who are acknowledged experts in this field and who vindicated the findings of the preliminary review. Therefore, traditional systematic reviews, that take an aggregative approach to summarise data, would not do justice to the depth and breadth of available evidence. This indicated that an alternative approach, which incorporates some elements of a systematic review but which allows for interpretation and quality appraisal, would be more suitable.

Furthermore, CIS allows for the generation of practice-based theory which is compatible with the objective of the review to develop a theoretically informed understanding of the phenomenon in question.

This protocol has been registered at the International Prospective Register of Systematic Reviews (PROSPERO) (registration number: CRD42024542929).

## Searching the literature

**Eligibility criteria.** While conventional systematic reviews aim to identify and include all relevant literature meeting fixed inclusion criteria, interpretive reviews set out to identify potentially relevant papers close to the topic of interest [18]. Though boundaries are less well-defined, there is still a requirement to identify some eligibility criteria to limit the literature to be included.

The criteria for inclusion in the review will be as follows:

- Population—adults (aged 18 years and over) with an intellectual disability and/or those providing care (family, formal and/ or informal caregivers)

- Language—English

- Study design–any

- Publication type–all types including primary research, reviews, reports, commentaries, editorials and position papers

- Content–must contain reference to care at end-of-life for people with an intellectual disability

- Timeframe—no restrictions

As the review progresses, it may be necessary to refine the eligibility criteria to ensure that the insights emerging from the review are incorporated.

**Search strategy.** A broad search strategy, based on the key concepts of 'care at end-of-life', 'people with an intellectual disability' and 'end-of-life' was developed in consultation with subject librarians. The search will utilise identified keywords with a combination of database specific control language. Similar search strings will be performed across five electronic databases—CINAHL, EMBASE, MEDLINE, PsycINFO and Web of Science—resulting in a combined database of all search results. Minor adjustments will be made for each database to ensure optimisation. An example of the search string to be used for the EMBASE database is shown in Table 1.

**Literature screening.** A pilot of the initial search strategy will be carried out by MH and MMcM who will randomly sample 20 abstracts and screen for eligibility. Any required alterations to the eligibility criteria will be made following this initial assessment. Database searches will then be carried out and all papers identified will be imported into EndNote X21 for deduplication. The resultant papers will be exported to Covidence for title and abstract screening for eligibility by MH and MMcM. To assist in this stage of the screening process, the reviewers will refer to exclusion criteria, namely where the focus of the literature is on any of the following topics:

- The development and/or evaluation of end-of-life initiatives or resources for people with an intellectual disability

- The impact on carers of providing end-of-life care to people with an intellectual disability

**Table 1. Search string for EMBASE database.**

| Concepts | Search No. | Search Strings |
|---|---|---|
| **Concept 1: Care at end-of-life** | #1 | 'End of life care':ab,ti OR 'palliative care':ab,ti OR 'hospice care':ab,ti OR 'terminal care':ab,ti OR 'palliative therapy':ab,ti |
| | #2 | 'terminal care'/exp OR 'palliative therapy'/exp OR 'hospice care'/exp |
| | #3 | #1 OR #2 |
| **Concept 2: People with an intellectual disability** | #4 | 'Intellectual disabil*':ab,ti OR 'intellectually disabled':ab,ti OR 'learning disabilit*':ab,ti OR 'mental handicap*':ab,ti OR 'mentally handicapped':ab,ti OR 'mental retard*':ab,ti OR 'mentally retarded':ab,ti OR 'mental disabil*':ab,ti OR 'mentally disabled':ab,ti OR 'mental defici*':ab,ti OR 'mentally deficient':ab,ti OR 'mental impairment*':ab,ti OR 'mentally impaired':ab,ti OR 'intellectual impairment*':ab,ti OR 'intellectually impaired':ab,ti OR 'intellectual defici*':ab,ti OR 'intellectually deficient':ab,ti OR 'Down syndrome*':ab,ti OR 'Intellectual Development Disorder':ab,ti |
| | #5 | 'mental deficiency'/exp OR 'learning disorder'/exp |
| | #6 | #4 OR #5 |
| **Concept 3: End-of-life** | #7 | 'End of life':ab,ti OR 'death':ab,ti OR 'dying':ab,ti OR 'terminal':ab,ti OR 'life limited':ab,ti |
| | #8 | 'death'/exp OR 'dying'/exp |
| | #9 | #7 OR #8 |
| **Combined** | #10 | #3 AND #6 AND #9 |

Other sources will be explored for grey literature. This will involve searching supplementary databases including Cochrane Library, Global Index Medicus (WHO), Google Scholar and Proquest Dissertations and Theses. Additional strategies will be used to identify other potentially eligible papers, such as reviewing reference lists of selected publications and focussed searches of relevant websites.

- Decisions around medical interventions to people with an intellectual disability at end-of-life

- Including people with an intellectual disability in end-of-life research

- Culturally specific issues relating to end-of-life care for people with an intellectual disability

- The impact on people with an intellectual disability of the end-of-life of others

Papers identified as 'eligible', classified as either meeting the inclusion/exclusion criteria or unclear whether they should be included, or where the abstract is not available, will subsequently undergo full-text screening.

Any uncertainty regarding study inclusion will be resolved by consultation and, where disagreements continue, a third reviewer will mediate. This referral procedure will be carried out should ambiguities emerge at any stage of the review process.

## Determination of quality

The full-text of all non-excluded papers will be retrieved and reviewed independently by each reviewer to make an assessment of whether they are relevant to the topic under review.

As in the original [18] and subsequent CIS reports [19–21], the preliminary step in the quality appraisal phase will be to identify papers classed as 'fatally flawed'. The criteria to be used will apply to all empirical papers (excluding reviews) and will be as follows—(1) the aims and objectives of the research are clearly stated; (2) the research design is clearly specified and appropriate for the aims and objectives of the research; (3) a clear account is provided of the

process by which findings were reproduced; (4) sufficient data is displayed to support interpretations and conclusions and; (5) the method of analysis is appropriate and adequately explicated.

Following the exclusion of empirical papers that are found to be 'fatally flawed' according to this criteria, the relevance of each remaining paper will be appraised. A high tolerance threshold will be applied when appraising the remaining papers to maximise the inclusion of a wide range of literature. The objective of the appraisal process will be to prioritise papers based on their ability to generate concepts, theories or insights in relation to the question under review [18]. The following broad question will be used to guide this stage of the appraisal process: Does the paper provide clear insights into what is good quality end-of-life care for people with an intellectual disability?

MH, PP and MMcM will follow the steps outlined above to appraise each paper. Papers which meet the criteria outlined above and are therefore deemed by consensus to be 'relevant' will be included in the sample frame of literature.

## Sampling

If, as in the case of Bullock et al. [19], the number of relevant papers is excessive, a purposive sample–identified from the quality appraisal as having a clear focus on end-of-life care—will be drawn from this sample frame of literature for the synthesis [22]. Sampling is justified as the focus in an interpretive synthesis is on the development of concepts and theory rather than on an exhaustive summary of all data [18]. Alternatively, if the volume of research identified following critical appraisal is relatively small, all relevant papers will be included in the synthesis rather than a sample from the larger pool [23].

## Data extraction

The following data will be extracted from all papers—title, publication year, author name, country of focus and publication type. For empirical research, the following details will also be extracted -study design, method of data collection, type and number of research participants. Key findings from each paper will be summarised.

An additional step will be to conceptually map data from the findings section of each paper according to the four characteristics of excellence in palliative and end-of-life care provision for people with an intellectual disability, as identified by Tuffrey-Wijne and Davidson [17]. These are individual and organisational commitment; working together in collaboration; the person's story is at the heart of care and developing tools and staff training Additional data, not reflected in any of the framework themes but worthy of inclusion, will also be extracted. This conceptual mapping exercise will help categorise the data into topics of interest and facilitate the subsequent analysis and synthesis. The standardised form to be used for data extraction is available in S1 Table.

Data will be extracted by MH who will ensure accuracy in relation to the quantitative data extracted and completeness in relation to the qualitative data. The data extraction process will involve critical discussion between all authors to ensure a line of argument is developing that will feed into the final synthesis.

## Data analysis and synthesis

The analysis, synthesis and integration processes will be iteratively conducted and involve ongoing discussion between reviewers [22]. As recommended by Dixon-Woods et al., the credibility of the evidence will be critically examined throughout to evaluate the contribution being made to the development of the synthesising argument.

Principles from a "best-fit" framework approach [24] will be adopted to analyse the findings and ultimately produce a synthesising argument. Using the identified framework [17] as the initial basis, data extracted from each paper will be integrated and coded against pre-existing themes. New themes will be generated, if necessary, in response to the evidence reported in the data. In-depth examination of the data will enable common themes and concepts to be identified. Constant comparison of the theoretical structures being developed in response to the data will be required as theoretical constructs, based on the emerging themes and concepts, will be developed [18]. During this process, a synthesising argument will emerge in the form of a new theoretical framework outlining the key characteristics in the provision of care that are required to ensure that people with an intellectual disability have a good end-of-life experience.

The developing theoretical framework will guide further selection of relevant literature, if required, to fill any conceptual gaps identified. These gaps will be filled by conducting additional purposive sampling from the sampling frame or by conducting additional purposive searches, adapting the initial inclusion criteria if necessary [23]. Evidence from the literature will continue to be integrated into the developing theoretical framework until saturation is reached [22].

## Discussion

This protocol sets out how a CIS approach will be used to synthesise a diverse body of evidence concerning end-of-life care for people with an intellectual disability. A key strength of this approach is that it will allow for the selection of a theoretically rich sample of literature which is appropriate for a study of this nature with highly heterogeneous literature. A further advantage of CIS is that it will allow for a critical approach to existing evidence, facilitating theory generation and allowing new insights to emerge [25] through an iterative, dynamic and critical synthesis of the literature [18]. This will provide an opportunity to challenge—against research-based evidence—an existing framework [17] which was developed based on examples of good practice. A potential limitation of adopting a CIS approach is the suggestion that there may be a lack of transparency [26], particularly in relation to the methods used to generate the synthesising argument. This will be addressed by clearly outlining how these stages of the review will be conducted and by making available the data which will be drawn from the literature and integrated into a synthesising argument.

The findings from this review will be submitted for publication in an Open Access health journal and will be presented at local, national and international events and conferences.

## Conclusion

This protocol describes the process for conducting a CIS review to develop a theoretical framework outlining the key characteristics in the provision of care that are required for people with an intellectual disability to have a good end-of-life experience. It is believed there is a need for a comprehensive, systematic, well-organised theoretical framework to support the implementation of best practice in this area. It is anticipated that this will help address the inequities that people with an intellectual disability experience accessing health care that extends to end-of-life care.

## Supporting information

**S1 Table. Data extraction form.**
(DOCX)

## Acknowledgments

The authors would like to acknowledge subject librarians Jessica Eustace Cooke and Jessica Leonard for their expertise in assisting to create the search strategy.

## Author Contributions

**Conceptualization:** Margaret Haigh, Mary McCarron, Philip McCallion, Martin McMahon.

**Data curation:** Margaret Haigh.

**Investigation:** Margaret Haigh.

**Methodology:** Margaret Haigh, Martin McMahon.

**Project administration:** Margaret Haigh, Mary McCarron, Philip McCallion, Martin McMahon.

**Supervision:** Mary McCarron, Philip McCallion, Martin McMahon.

**Writing – original draft:** Margaret Haigh.

**Writing – review & editing:** Margaret Haigh, Mary McCarron, Philip McCallion, Pavithra Pavithra, Martin McMahon.

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
