## [Decision Letter · Decision Letter 0]

23 Jul 2024

PONE-D-24-17995Good quality end-of life care for people with an intellectual disability: a critical interpretive synthesis protocolPLOS ONE

Dear Dr. Haigh,

Thank you for submitting your manuscript to PLOS ONE. After careful consideration, we feel that it has merit but does not fully meet PLOS ONE’s publication criteria as it currently stands. Therefore, we invite you to submit a revised version of the manuscript that addresses the points raised during the review process.

We look forward to receiving your revised manuscript.

Kind regards,

Mickael Essouma, M. D.

Academic Editor

PLOS ONE

2. In the online submission form, you indicated that [No datasets were generated or analysed during the current study. All relevant data from this study will be made available upon study completion.]. 

Additional Editor Comments:

MY EDITORIAL COMMENTS ABOUT PONE-D-24-17995

1. Consider conforming to PLOS ONE guidelines for formatting manuscripts and go to PLOS ONE website to look at published manuscripts in the journal. The abstract must be in one block without subheadings. Consider deleting the “Protocol aim” sub-section and the information reported in that sub-section must be in the last paragraph of the introduction. Consider including keywords in the manuscript after the abstract section. Those keywords should be among the important words in the title and the manuscript that will facilitate the retrieval of your future published paper online. Format references according to the journal’s policy. Do you not see that there are way more references that would be expected in a protocol article?

2. Title: are you going to keep “Good quality end of life care” or will you revise it to “High/Best/Optimal quality end of life care”? Because we usually talk of high-quality care, best-quality care, and optimal-quality care. Furthermore, you state in the introduction that the definition of “good end-of-life care” for this population group is still a matter of debate.

3. The current introduction section is very long and does not clearly help the reader understand why you are undertaking this study, especially because you will generate theoretical frameworks. The introduction needs to be reduced to max 1.5 pages (4 paragraphs max). I propose the report of these bits of information: what is the definition of intellectual disability? What is their best classification? Highlight the multiple health challenges encountered by individuals with intellectual disabilities (and their potential sociologic determinants such as diverse forms of inequalities/discrimination) that lead to their reduced quality of life and life expectancy compared to the general population. You could go further by saying that although their life expectancy appears to be lower than that of the general population, it is increasing, which could lead you to think more and more about their health, the quality of care they receive and their quality of life when they are adults and even elderly. Regarding quality of care, it would be more important to focus on the quality of end-of-life care to help them end their lives with dignity and respect. Finally, you need to clearly state your research objectives.

4. Methods section: The information on lines 92-121 is confusing and somewhat unnecessary. I suggest revising that text by just stating that you will conduct and report this critical interpretive synthesis based on standard methods, and then you add current reference 30 which is indeed a good reference for the reporting of methods in this study. Then, by mentioning the different recommended sections of a critical interpretive synthesis article as sub-sections of the methods section in your article, unaware readers will understand by themselves how methods of a critical interpretive synthesis are reported. They will be able to find explanations in current reference 30. This means I do not agree with yyour sentence on lines 122 and 123. Consider conforming to reference 30 for writing the methods section of this article (please go through current reference 30 again to grasp the specifics needed in each sub-section), with the following methods’s sub-sections: searching the literature (information sources and search strategies), sampling (study selection), determination of quality (quality appraisal: conform to the quality appraisal of a critical interpretive synthesis, which is somewhat different from that of systematic reviews), data extraction, critical interpretive synthesis (reciprocal translational analysis, refutational synthesis, lines-of-argument synthesis). Regarding the “searching literature” sub-section, … Why did you choose to only describe the search ctrategy in EMBASE? Why not for MEDLINE (PUBMED)? Could you report the different strategies in search databases as in this article (doi: 10.1371/journal.pone.0305112)? What is the search period? Please, make a thorough description of the critical interpretive synthesis sub-section. Propositions in the line-of-argument synthesis part of the critical interpretive synthesis sub-section should be based on your current knowledge about the quality of end-of-life care for people with intellectual disabilities which will be briefly highlighted. Go through current reference 30 and these two reviews for further inspiration https://doi.org/10.1016/j.contraception.2021.11.004 and https://doi.org/10.1093/heapol/czae030.

5. Discussion section: consider adding limitations statements. For example, your propositions in the lines-of-argument synthesis part of the CIS sub-section of the methods in this protocol article could be different from those in (or less elaborated than in) the final articles of the research as you may make new discoveries while conducting the study. This should be acknowledged herein. And delete the dissemination plan section. Rather mention how you plan to disseminate results of your research at the end of the abstract and the discussion section without explicitly stating that you are reporting the dissemination plan.

6. Consider adding a conclusion section after the discussion section.

7. Declarations section should come after the conclusion section (except for the Funding statement and data availability statement which is required above as you did):

7.1. Funding: I am curious to know why your funder funded the study if they had no involvement in the research up to the decision to submit the article for publication.

7.2. Data availability statement: did yo want to state that no dataset was generated or analyzed for this protocol article?

7.3. Where is the statement about conflicts of interests?

7.4. Acknowledgments not reported.

8. English language editing will be necessary so that the manuscript is as short as possible. I suggest deleting the figure in the current manuscript because I do not see its relevance.

Reviewers' comments:

Reviewer's Responses to Questions

**Comments to the Author**

1. Does the manuscript provide a valid rationale for the proposed study, with clearly identified and justified research questions?

Reviewer #1: Yes

2. Is the protocol technically sound and planned in a manner that will lead to a meaningful outcome and allow testing the stated hypotheses?

Reviewer #1: Yes

3. Is the methodology feasible and described in sufficient detail to allow the work to be replicable?

Reviewer #1: Yes

4. Have the authors described where all data underlying the findings will be made available when the study is complete?

Reviewer #1: Yes

5. Is the manuscript presented in an intelligible fashion and written in standard English?

Reviewer #1: Yes

6. Review Comments to the Author

You may also provide optional suggestions and comments to authors that they might find helpful in planning their study.

Reviewer #1: This protocol submission comprehensively describes the methods to be utilized to undertake a critical interpretive synthesis on a significantly important topic area.

7. PLOS authors have the option to publish the peer review history of their article (what does this mean?). If published, this will include your full peer review and any attached files.

Reviewer #1: No

---

## [Author Response · Author response to Decision Letter 0]

3 Sep 2024

Please refer to detailed rebuttal letter included with this re-submission.

---

## [Editor Report · Decision Letter 1]

4 Sep 2024

PONE-D-24-17995R1Good quality end-of life care for people with an intellectual disability: a critical interpretive synthesis protocolPLOS ONE

Dear Dr. Haigh,

Thank you for submitting your manuscript to PLOS ONE. After careful consideration, we feel that it has merit but does not fully meet PLOS ONE’s publication criteria as it currently stands. Therefore, we invite you to submit a revised version of the manuscript that addresses the points raised during the review process.

We look forward to receiving your revised manuscript.

Kind regards,

Mickael Essouma, M. D.

Academic Editor

PLOS ONE

Journal Requirements:

Additional Editor Comments:

You have already stated in the introduction and at the beginning of the methods section that this is the protocol of a critical interpretive synthesis. You do not need to repeat in all sub-sections of the methods section and elsewhere in the manuscript "in a critical interpretive synthesis" and what a criticial interpretive synthesis is about. For example, the sentence on lines 172 to 174 is unnecessary.

Avoid bullet as much as possible in the text, and make sentences. Example on lines 182 to 185. Along this line, reduce the text under the data analysis sub-section of the methods section with only information about how you will synthesize and interpret data in your review.

Please, can you add links of websites where references 5, 13 and 16 can be found?

Line 49: I would say "It was" because the systematic review suggesting 1% was published in 2011 and the frequency may actually have changed.

Lines 81-83: "This decision was informed by the findings of an initial review of the literature in relation to the end-of-life experiences of people with an intellectual disability which established that the body of evidence is similarly diverse and complex." The reference supporting this statement (of the corresponding systematic review) is needed.

Lines 89 and 90: "This protocol will use the Preferred Reporting Items for Systematic 89 Review and Meta-Analysis Protocols (PRISMA-P) 2015 checklist [19, 20] (S1 Table)." Consider deleting this sentence and S1 Table because this is not the protocol of a systematic review whilst PRISMA-P guides reporting of systematic review protocols.

Under the "Searching the literature" sub-section of the Methods section, consider presenting the search strategy (specifying the search period as well) before eligibility criteria, and then literature screening. Consider revising the eligibility criteria. Inclusion criteria would best be framed indicating study designs ( I guess you will find quantitative observational and interventional studies, qualitative and mixed-methods studies) that will be included as well as using the PICO (Population [this should be only individuals with intellectual disability, not care providers as well as you report in the present manuscript], Intervention [End-of-life care what specification of what you mean by end-of-life care as I mentioned in the last round of review], Comparator [Individuals without intellectual disabilities for studies using a comparator group], and Outcome [clearly define the outcomes you will be looking for: quality of life...]. Exclusion criteria are not teh contrary of inclusion criteria, but criteria that led to the exclusion of studies after their initial fulfilment of inclusion criteria.

Line 139: did you want to say "Any uncertainty regarding study inclusion"?

Line 143: low threshold of what: study quality? The quote should be removed on eligible.

The questions on lines 150 to 155 should be on the same line following the text on line 149. Is there a reference for the criteria listed on lines 148 to 155? Consider adding it. Altogether, there is some confusion about how you will determine the quality of studies to include in your review, making it necessary to clearly revise the text on lines 143 to 162.

Consider revising the text under the data extraction sub-section reducing its length with only the information about the data that will be extracted (consider extracting data about the study [design, author, location, method of data collection, study period] and PICO elements because those are elements that will help you meet your review goals), how you will extract those data and how you will resolve discrepancies arising between people who will extract data.

Line 224: is the word "complex" necessary here since everything can be considered to be complex?

Lines 225 to 227: "The approach will entail adopting procedures of conventional systematic review methodology to search and review the literature, combined with interpretive synthesis associated with qualitative research." I do not agree with this sentence. Consider sticking to the critical interpretive synthesis although we know that this type of review incorporates some elements of the systematic. It is at the beginning of the methods section that you should showcase how critical interpretive synthesis incorporates elements of the systematic review and not repeat that description elsewhere in the manuscript. If you really want to extensively describe how a critical interpretive synthesis compares with a systematic review, you could do that in a table that you append to the text at the beginning of the methods section.

Why has one author been added? And the reasons of changes in authors positions should be clearly mentioned.

---

## [Author Response · Author response to Decision Letter 1]

15 Sep 2024

Please refer to rebuttal letter for responses to editor comments.

---

## [Editor Report · Decision Letter 2]

23 Sep 2024

Good quality end-of life care for people with an intellectual disability: a critical interpretive synthesis protocol

PONE-D-24-17995R2

Dear Dr. Haigh,

We’re pleased to inform you that your manuscript has been judged scientifically suitable for publication and will be formally accepted for publication once it meets all outstanding technical requirements.

Kind regards,

Mickael Essouma, M. D.

Academic Editor

PLOS ONE

Additional Editor Comments (optional):

The manuscript has been improved satisfactorily.

Last few comments.

Consider backing the sentence “This decision was informed by the findings of an initial review of the literature in relation to the end-of-life experiences of people with an intellectual disability which established that the body of evidence is similarly diverse and complex.” on lines 77-79 with this sentence "This was a preliminary review carried out by the lead author and involved semi-structured searches of electronic databases. Discussions were also held with members of the research team who are acknowledged experts in this field and who vindicated the findings of the preliminary review." from the rebuttal letter.

Consistency with references cited in the text and in the reference list is needed. For instance, the work by Bullock et al is 21 in the text (line 164), but 19 in the reference list. On line 179, a reference 28 is cited whereas the reference list contains 26 references.

On line 159, "what is" appears twice.
---

## [Editor Report · Acceptance letter]

29 Oct 2024

PONE-D-24-17995R2 

PLOS ONE

Dear Dr. Haigh, 

I'm pleased to inform you that your manuscript has been deemed suitable for publication in PLOS ONE. Congratulations! Your manuscript is now being handed over to our production team.

Kind regards, 

on behalf of

Dr. Mickael Essouma 

Academic Editor

PLOS ONE